# Superwetting Polymeric Three Dimensional (3D) Porous Materials for Oil/Water Separation: A Review

**DOI:** 10.3390/polym11050806

**Published:** 2019-05-06

**Authors:** Yihao Guan, Fangqin Cheng, Zihe Pan

**Affiliations:** 1Institute of Resources and Environmental Engineering, Shanxi University, 92 Wucheng Road, Xiaodian District, Taiyuan 030006, China; guanyihao007@163.com (Y.G.); cfangqin@sxu.edu.cn (F.C.); 2Shanxi Collaborative Innovation Center of High Value-Added Utilization of Coal-Related Wastes, Shanxi University, Taiyuan 030006, China

**Keywords:** porous polymer materials, superhydrophobic, superoleophobic, superhydrophilic, oil/water separation

## Abstract

Oil spills and the emission of oily wastewater have triggered serious water pollution and environment problems. Effectively separating oil and water is a world-wide challenge and extensive efforts have been made to solve this issue. Interfacial super-wetting separation materials e.g., sponge, foams, and aerogels with high porosity tunable pore structures, are regarded as effective media to selectively remove oil and water. This review article reports the latest progress of polymeric three dimensional porous materials (3D-PMs) with super wettability to separate oil/water mixtures. The theories on developing super-wetting porous surfaces and the effects of wettability on oil/water separation have been discussed. The typical 3D porous structures (e.g., sponge, foam, and aerogel), commonly used polymers, and the most reported techniques involved in developing desired porous networks have been reviewed. The performances of 3D-PMs such as oil/water separation efficiency, elasticity, and mechanical stability are discussed. Additionally, the current challenges in the fabrication and long-term operation of super-wetting 3D-PMs in oil/water separation have also been introduced.

## 1. Introduction

The rapid development of textile, steel, metallurgy, petrochemical, and other industries has produced a large amount of oily wastewater, causing serious pollution to rivers, lakes, and other water bodies [1,2,3,4]. Among which, oil spillage during mining, transportation, and storage in the petroleum industry are the primary sources of oily wastewater emission [5,6]. For instance, the rapid spreading of the oil slick and the emulsified oil that occurred during mining in the Gulf of Mexico in 2010 (Figure 1) [7,8] and the Bohai Bay in 2011 [9] have caused a devastating blow to marine life and brought disasters to the ecosystem. The spilled oil floated on sea (Figure 1a) [10] threatens the living of sea creatures e.g., oil covered turtle (Figure 1c) [11] and brown pelican (Figure 1d) [12], and trapped them in the spilled oils preventing their swimming or flying while the burned oil released heavy smoke, which caused serious air pollution (Figure 1b) [13]. Oil spills to water not only threats the safety of the ecosystem and human society but also is a great waste of resources [3]. For this world-wide environmental problem, scientists are constantly exploring new and effective technologies and materials to separate oil/water mixtures and recycle oil from oily wastewater efficiently.

Traditional methods including gravity-driven separation, filtration, flocculation, flotation, adsorption, biodegradation, etc. [14,15,16] have been widely utilized to purify oily wastewater. These methods show unique advantages in oily wastewater treatment. Gravity-driven separation can effectively separate oil from water by using the density difference between oil and water [17]. Traditional membrane filtration allows the penetration of small water molecules while prohibiting the passing through of oils under the assistance of pressure or electrical potential achieving oil/water separation [18,19]. Flocculants are commonly used agents to eliminate floated, dispersed, and emulsified oils in water through coagulating oils into large precipitates to achieve oil/water separation [20]. Floatation separates the oil/water mixtures via pushing oil droplets to the surface by bubbling [21]. The mixed oils in water can also be removed by adsorbing. Many adsorbents e.g., wood sheets, zeolites, fabrics, cotton, and linen etc. with large surface area and porosity are easy to obtain from nature and widely used for oil adsorption to separate oil from water [22,23]. Small amounts of oils can be biodegraded into small molecule or CO_2_ and water by the microorganisms in water to purify oily waste water [24,25]. Nevertheless, the significant shortcomings of these methods, for instance, low separation efficiency and large space occupation of gravity separation, fouling and poor oil/water selectivity of membrane separation, difficulty in reuse/recycling of adsorbents and flocculants, low adsorption capacity and secondary pollution, etc., restrict their practical applications in oily wastewater treatment [6,26,27,28].

Since 1997, the German botanist Baethlott [31] reported that water droplets sit on lotus leaves with water contact angle larger than 150° and roll off easily carrying away the dust when tilting and this phenomenon is defined as the “lotus effect”. This extraordinary property is named as superhydrophobicity and self-cleaning [32,33,34]. Further investigation showed that lotus leaf is comprised of nano/microhierarchical structures and they are coated with low surface energy wax to reduce surface energy, thereby delivery superhydrophobicity and self-cleaning property (Figure 2a) [29,30]. This unique property can be used to design superhydrophobic membranes to eliminate oil from water achieving oil/water separation. In 2004, Feng et al. [35] developed superhydrophobic-superoleophilic copper mesh by dip-coating hydrophobic polytetrafluoroethylene and oil was removed from water through this membrane under gravity-driven separation. Another example learning from nature is mimicking the superhydrophilic-underwater superoleophobic structures from fish scales (regular fan-shaped microstructures (Figure 2b)) to fabricate oil/water separation membranes to remove water from oil/water mixture and keep cleaning from oils [36]. In 2011 Xue et al. [36] fabricated superhydrophilic polyacrylamide (PAM) mesh, which allows water penetration while repelling oil, thereby reduces oil pollution. Dunderdale et al. [37] prepared a more advanced amphiphilic stainless steel mesh which was modified by hydrophilic polymethacrylic acid (PMAA) and hydrophobic poly(ethylhexylmethacrylate)(PEHMA) and can switch its surface wettability in response to various stimulations to effectively separate oil/water mixtures. After that, oil/water separation material with biomimetic super-wettability has undergone rapid development [38,39]. However, the issues of membrane fouling and weak intrusion pressure restrict the long-term operation. Recently, two meshes with opposite wettability were modified by polymer brush and combined to treat oil/water solutions, resulting in high efficiency under high flux and keeping surfaces free from oil contamination [40,41]. Nevertheless, membranes cannot be used to adsorb or store the spilled oils. Therefore, designing a new type of oil/water separation material with high selectivity, adsorption capacity, anti-fouling property, and recyclability is essential for long–term oily wastewater treatment.

To increase the penetration channel and separation efficiency, there are some attempts at using inorganic and metal-based 3D-PMs (normally are ceramic foams or copper/nickel foams etc.) [42,43,44,45]. However, their significant disadvantages such as brittle, lack of elasticity, ease of corrosion in acid/base environment, unable to be reused, and difficulty in surface modification restrict the application in oily wastewater treatment. Recently, polymer-based three-dimensional porous materials (3D-PMs) (such as sponge [46,47,48], foam [49,50,51,52], and aerogel [53,54,55]) with multi-layered network structure, higher porosity, adsorption capability, long penetration channel, and tunable pore structure are regarded as promising alternates for effectively oil/water separation after being functionalized with special wettability [56,57,58,59,60]. This is because polymers are always light in weight, ease of processability, resistance to corrosion, chemical stability, and the abundant functional groups on the surface which can be modified with a variety of materials easily [53,61,62,63,64,65]. Especially, the excellent flexibility of polymers enables the high elasticity of polymer 3D-PMs. The elastic 3D-PMs can be compressed and recycled under a certain pressure to discharge the adsorbed oil by compression, which simplifies the separation process [65,66]. Therefore, polymers have been widely used in developing 3D-PMs and polymeric 3D-PMs have become one of the research hotspots in oil/water separation. The published articles on superwetting polymeric oil/water separation 3D-PMs (from 2011 to February 2019) are shown in Figure 3, indicating that polymeric 3D-PMs have undergone rapid development. Interfacial superwetting 3D-PMs are mainly divided into four categories according to their surface wettability and their functions in dealing with different types of oil/water mixtures: (i) superhydrophobic-superoleophilic 3D-PMs are used to absorb oil; (ii) superhydrophilic-underwater superoleophobic 3D-PMs are used to eliminate the layered oil/water mixture and oil-in-water emulsions; (iii) superhydrophilic-superoleophobic 3D-PMs are used to remove water from oily wastewater and eliminate water from water-in-oil emulsion; (iv) 3D-PMs with switchable super-wettability, which can change wettability and remove oil or water by responding to the external stimulation. In this feature article, firstly, the theoretical mechanism of oil/water separation of super-wetting 3D-PMs is reviewed. Then, the design routes, techniques, the applications in different oil/water mixtures treatment, the current challenges existed in 3D-PMs oil/water separation such as oil contamination, relatively weak mechanical robust under certain pressure, difficulties in oil-desorption, and the adsorption or separation efficiency, were also discussed. Finally, the perspective of the future development on fabrication 3D-PMs to meet the practical requirements is briefly discussed.

## 2. Theoretical Background

Surface wettability is the fundamental property of 3D-PMs for the applications in oil/water separation. Either hydrophilic or hydrophobic 3D-PMs cannot be directly used because both oil and water can wet and pass through the porous structure [62,67,68,69]. Thus, developing the 3D-PMs with selectively oil/water separation property is critical and can be achieved by creating special wettability [60]. The classic theories of Wenzel [70] and Cassie–Baxter indicated that surface wetting is dominated by the chemical composition and surface structure [33]. Thus, the modification of surface energy and the construction of proper pore structure are vital to super-wetting 3D-PMs. Besides, the wettability and surface structure are also an important parameter to determine the separation efficiency and the surface fouling.

### 2.1. Wetting on Porous Surface

#### 2.1.1. Contact Angle in Air

Wetting behaviors on solid surface is mostly reflected by measuring the water contact angle (WCA) [71]. For an ideal smooth surface, WCA refers to the results of interactions between gas, liquid, and solid phases (Figure 4). Usually, the WCA can be obtained by Young’s equation [72]:
(1)$$\cos\theta =\frac{\gamma_{\mathrm{gs}}-\gamma_{\mathrm{sl}}}{\gamma_{\mathrm{gl}}}$$
where *θ* represents the intrinsic contact angle, and *γ*_gs_, *γ*_sl_, *γ*_gl_ are the surface energy of gas-solid, solid-liquid, and gas-liquid interfaces, respectively. When *θ* < 90°, the surface is hydrophilic (Figure 4) while the surface is hydrophobic when *θ* > 90° (Figure 4). 

Since the surface of 3D-PMs is composed of micro/nanopores and is not smooth [74], its wettability cannot be obtained directly from Young’s equation [70,75]. The wetting behaviors of water on porous or rough surfaces can be illustrated by Wenzel and Cassie–Baxter theories [70,76] that the wettability of real surface is the synergetic effects of the surface energy and the surface structure. Wenzel systematically investigated the effects of roughness on wettability, showing that the wettability can be enlarged by surface roughness. In this model, water contacts the solid surface and completely wets and fills into the rough structure resulting in the larger actual contact compared to the apparent contact of the smooth surface (Figure 5a). Therefore, WCA can be calculated from Wenzel equation [70]:
(2)$$\cos\theta_{w}^{*}=r\;\cos\theta$$


In which *θ**_W_ and *θ* are the apparent contact angle and contact angle on ideal flat surface, respectively; *r* is the ratio of the practical contact area of the solid-liquid interface to the ideal flat surface (*r* > 1). The Wenzel equation shows that surface roughness is important for the enhancement of the solid surface wettability [77]. However, extremely high roughness or regular porous structure results in cos *θ**_W_ larger than 1 or less than −1, which is mathematically unacceptable. To illustrate this phenomena, Cassie–Baxter proposed [76] that the trapped air-pockets under water droplets are considered as the superhydrophobic medium, which repels water wetting and penetration. Thus, the liquid–solid interface is replaced by the air-liquid interface resulting in superhydrophobicity (Figure 5b). The contact angle is defined as Equation (4) [76]:
(3)$$\cos\theta_{\mathrm{CB}}^{*}=\varphi_{s}\;\cos\theta +\varphi_{s}-1$$
where *φ*_s_ is the ratio between the contacted solid and liquid. *θ**_CB_ represents the apparent contact angle of composite surface. The Cassie–Baxter state is metastable and can easily transit into stable state by breaking the air-pockets and filling with water under. This process can occur by the stimulating of external factors such as pressure [78], electric field [79] etc., the liquid may overcome the energy barrier to transition from Cassie–Baxter state to Wenzel state (Figure 5c) [80,81,82].

#### 2.1.2. Contact Angle Hysteresis

In actual oil–water separation, low surface tension oils tend to adhere to the separation materials causing the surface contamination and the reduction of oil/water separation efficiency [29,71,75,84]. Thus, the self-cleaning performance is a great concern of super-wetting materials and is normally characterized by contact angle hysteresis (CAH) [71]. The intrinsic meaning of CAH is the interactions between liquid droplets and the contacted solid surface. CAH reflects the smallest energy that the droplet needs to overcome to slide off. The smaller the CAH the less energy the liquid droplet has to overcome. CAH is the result of the difference between advancing contact angle (adv, Figure 5d) and the receding contact angle (rec, Figure 5d), in which adv and rec are is the largest and smallest angle contact angle at the starting point of droplet sliding [29,71,75,84]. The tilting angle of the solid surface when the water droplet started to roll off is defined as sliding-off angle which illustrates the adhesion between and liquid droplet [75,84]. The CAH, adv, rec and sliding-off angle (Figure 5e) reflects the motion behavior of water droplets on a surface and the self-cleaning property of solid surface that the smaller the CAH and sliding-off angle the smaller adhesion of surface and better self-cleaning property.
(4)$$CAH=\theta_{Adv}-\theta_{Rec}$$


#### 2.1.3. Contact Angles in Water or Oil Phases

As illustrated above the liquid contact angle in air can be predicted from Yong’s equation, the contact angle of liquid a (*l*_a_) in liquid b (*l*_b_) cannot be calculated from Young’s equation. In order to explain the wetting behaviors of liquid a on solid surface in liquid b and calculate the liquid a contact angle under liquid b (*θ*_LaCA_), the following equation has been derived from Young’s equation [30,85,86]:
(5)$$\cos\theta_{L_{a}\mathrm{CA}}=\frac{\gamma_{l_{a}\text{-}g}\cos\theta_{a}-\gamma_{l_{b}\text{-}g}\cos\theta_{b}}{\gamma_{l_{a}\text{-}l_{b}}}$$
where *γ*_l_a_-g_, *γ*_l_b_-g_ and *γ*_l_a_-l_b__ are interface tensions of liquid a-gas, liquid b-gas and liquid a-liquid b interface, respectively. *θ*_a_ and *θ*_b_ are the contact angle of liquid a and liquid a in air, respectively. (Figure 6a)

If a surface is hydrophilic, the contact angle of oil and water are less than 90° and *θ*_water-air_ is larger than *θ*_oil-air_. Thus, the value of cos*θ*_oil-air_ and cos*θ*_water-air_ are positive and an oleophobic surface in water is reached when *γ*_oil-air_cos*θ*_oil-air_ is smaller than *γ*_water-air_cos*θ*_water-air_ (because the surface tension of water is larger than oil). When water contacts with solid surface in oil phase, cos*θ*_water-oil_ is positive and oil is replaced by water showing hydrophilicity in water. The value of cos*θ*_water-air_ and cos*θ*_oil-air_ of a hydrophobic-oleophilic surface in air are negative and positive, respectively, such surface shows underwater hydrophilicity and underoil hydrophobicity correspondingly which can be calculated from Equation (5) (Figure 6d,e). 

Cassie–Baxter mode is also available for a rough surface in water/oil environment and the wettability of the solid surface can be expressed as the following equation [30,76,88] (Figure 6b):
(6)$$\cos\theta_{L_{a}\mathrm{CA}}^{*}=\varphi_{s}\cos\theta_{L_{a}\mathrm{CA}}+\varphi_{s}-1$$
where *φ*_s_ is the ratio between the contacted solid and liquid a, *θ*_LaCA_ is the intrinsic liquid a contact angle in liquid b. In water phase, water molecules are trapped in the nano/microstructures and forms water-solid interface which prohibits the contact between oil and solid surface showing superoleophobicity (Figure 6c). 

### 2.2. Oil/Water Separation Theory Analysis

To better explain the oil/water separation behaviors of superwetting 3D-PMs and make the separation process more clearly, it is hypothesized that the surface or any section is idealized as a mesh plane according to the Yong-Laplace equation [74,89]. Usually, the surface of the liquid is horizontal which generates the external pressure on porous surface. Before the liquid completely wets and touches the bottom of pores, an intrusion pressure Δ*p*_it_ have to be overcome, which is described as [90,91,92,93]:
(7)$$\Delta p_{it}=\frac{2\gamma }{R}=-4\gamma (\cos\theta_{A})/d$$
where *γ* is interface tension of water/oil, *R* is the sedi-diameter of the meniscus, *θ*_A_ is advancing contact angle of the liquid, and *d* is the average diameter of the pore. Equation (7) shows that when *θ*_A_ < 90°, i.e., the intrusion pressure $\Delta p_{it}=p_{in}-p_{out}<0$, the intrusion pressure is broken and oil spontaneously permeates through the pores driven by gravity. Conversely, when *θ*_A_ > 90°, i.e., the intrusion pressure $\Delta p_{it}=p_{in}-p_{out}>0$, means that the hole can sustain pressure to some extent, and the water is blocked outside [92,93].

For a surface with *θ*_A,water_ > 150°, the water intrusion pressure Δ*p*_,water_ > 0 while the oil intrusion pressure Δ*p*_,oil_ < 0 (*θ*_A,oil_ < 30°) (Figure 7a,b) that water is prevented while the oil can easily break through the porous structure driven by the gravity achieving oil/water separation. On the contrary, for a porous surface with *θ*_A,water_ < 30° which is superhydrophilic allowing water passing through. This is because such superhydrophilic porous materials must be firstly wetted by water to form the water films [74] under capillary force thereby repel the penetration of oil pass through. However, in long-term operation the water film tends to move to the bottom of the hole and oil starts to fill the pores under the effect of gravity, causing the reduction of oil repellency. As time goes by, the water-solid interface in the pore is replaced by oil-air interface gradually resulting in the decrease of Δ*p*_,oil_ (Figure 7c) [89]. 

The oil/water separation efficiency of 3D-PMs can be evaluated by oil absorption capacity *k* and separation efficiency R following equations [94,95]:
(8)$$k\;\left(g/g\right)=\left(m_{e}-m_{o}\right)/m_{o}$$
where *m*_o_ is initial mass of porous material, *m*_e_ is the total mass of the porous material and the absorbed liquid [36].
(9)$$R\%=1-C_{p}∕C_{o}\times 100\%$$
where *C*_0_ and *C*_p_ are the oil concentrations in oily wastewater mixture before and after de-oiling.

## 3. Fabrication Special-Wettable Oil/Water Separation Three Dimensional Porous Materials (3D-PMs)

The fabrication of super-wetting 3D-PMs follows the classical theories by the combination of creating hierarchical structures (nano/microstructure) and surface chemistry modification. The surface structure can be fabricated via many methods such as etching, nanomaterials modification while the surface chemistry can be tubed through low or high surface energy molecules. During the past decades, a variety of research articles have been published on fabricating special wettable 3D-PMs through post-modification, self-assembly, phase separation etc. (Table 1). The post-modification is the technique to modify on commercial polyurethane (PU) [96], melamine foam [97], polymethyl methacrylate (PMMA) sponge/foam [51,98] with inorganic nanomaterial or inorganic-organic composite as modification agents through vapor deposition, dip-coating, chemical etching, and solution immersion. Self-assembly includes polymerization, freeze-drying, sol−gel process, electrospinning, phase separation, etc. and aerogel and sponge with ultralight, large specific surface area and porosity can be fabricated through this method. The common used techniques to develop different type of super-wetting 3D-PMs are summarized in Table 1. The following sections review the design routes, techniques and materials which are used in developing special wettable 3D-PMs, their performances in oil/water separation and the current challenges.

### 3.1. Superhydrophobic-Superoleophilic Oil/Water Separation 3D-PMs

3D-PMs with superhydrophobicity-superoleophilicity repel water while it can be wetted by oils allowing the passing through of oils to remove oil from water. Nowadays, three types of superhydrophobic-superoleophilic 3D-PMs e.g., sponge, foams and aerogels have been developed to adsorb light oil from water surface [67,99,122,123]. The adsorption capacity of oils can reach as high as around more than one hundred times of the original weight of 3D-PMs and the separation efficiency can reach to 99% [124]. Currently, two common ways are employed to prepare superhydrophobic 3D-PMs porous materials: (a), utilize the inherent pore structures of commercial PU/melamine/PMMA sponges to achieve superhydrophobicity through surface modification e.g., dip-coating [117], solution-immersion [125] or vapor deposition [126]; (b), use the polymer, graphene, polymer composite as precursor to construct hydrophobic porous 3D structures through self-assembly approach such as foaming technology [105], freeze-drying [127] or electrospinning [128] methods.

#### 3.1.1. Preparation Superhydrophobic 3D-PMs on Commercial Foam/Sponge

Commercial sponge/foams (polyurethane, melamine, polyethylene, polystyrene, poly(melamine formaldehyde)) are inexpensive porous materials with high porosity (typically > 95%), low density (<18 mg/cm^3^), regular skeleton, elasticity, high mechanical property and chemical stability which are widely used in industry and daily life such as, thermal insulation, sound absorption and shock absorption [67,125,140]. These outstanding properties of commercial sponge/foam enable they can be used for oil/separation after proper modifications. Generally, both water and oil can wet and pass through commercial sponges which is lack of selectivity and oil/water separation performances [112]. However, designing superhydrophobicity to repel water penetration via different types of materials including organic, inorganic, and composite agents by changing surface energy or surface structure is regarded as an easy and feasible way.

##### Organic Agents

Commercial sponges/foams have abundant microporous structures and can reach superhydrophobicity after surface energy reduction and surface roughness increment. During the past decades, a variety of organic polymers with low surface energy such as hydrophobic organosilanes [108,125] and long-chain alkanes (i.e., fluorine/amine/thiol) [56,141] are extensively used to modify commercial sponges/foams by dip coating, solution-immersion, vapor deposition, situ-polymerization, chemical bonding, and crosslinking (Table 2). Zhang et al. [124] firstly etched the original polyurethane (PU) foam by CrO_3_ and H_2_SO_4_ solution to create rough structures on skeleton. The water contact angle on the sponge decreased from 56° to 0° and the sponge became superhydrophilc after etching. Then the hydrophobic fluoroalkylsilane (FAS) was grafted onto the sponge to reduce the surface energy and the water contact angle was larger than 155° with the sliding-off angle less than 5° via solution immersion. The self-cleaning property is very important for the foam and can be used to remove dust from the foam surface avoiding the blocking of pores (Figure 8e). The obtained superhydrophobic foam can selectively eliminate gasoline, crude oil, and petroleum ether from oily wastewater and the separating efficiency is above 95%. Methyltrichlorosilane with low surface energy was also reported to modify the PU sponge to reduce surface energy to delivery superhydrophobicity (Figure 8b) [108]. However, fluorosilane and chlorosilanes are usually toxic, with ease of deliquescent and hydrolysis to generate hydrogen fluoride or hydrogen chloride causing adverse effects to the environment. Organosiloxane is non-toxic organic matter with hydrophobicity, which can anchor to the surface firmly through cross-linking or situ-polymerization. Chen et al. [125] dip-coated the diluted polydimethylsiloxane (PDMS) and hexane precursor onto a melamine sponge to fabricate superhydrophobic sponge (WCA > 150°) for oil/water separation. This superhydrophobic sponge can absorb 45–75 times oils or organic solvents of its initial weight and maintains stable oil adsorption capacity after 20 cycles of oil absorption and squeezing. Xiong et al. [140] grafted the hydrolyzed 3-methacryloxypropyltrimethoxysilane (KH-570) onto PU foam through the reaction between alkoxysilane molecules and the isocyanate groups on PU foam and reached superhydrophobicity, which can absorb about 100 times its initial weight oil (lubricating oil, diesel oil, crude oil etc.) from oil/water mixture showing excellent oil adsorption property (Figure 8d). To increase the stability of the coated organic coatings, chemical bonding is regarded as an effective way. Oribayo et al. [57] firstly coated dopamine onto melamine sponge to form PDA coating via self-polymerization of PDA. Strong covalent and noncovalent bonds are formed during the polymerization of dopamine, which increased the durability of the coating resulting in long term operation. The superhydrophobicity of PDA-coated melamine sponge was achieved via grafting with low surface tension 1H, 1H, 2H, 2H perfluorodecanethiol (PFD) or octadecylamine (ODA) agents. PFD and ODA are chemically bonded to PDA surface through Michael or Schiff-base reaction via dipping-coating method [142]. The as-prepared superhydrophobic sponge can absorb about 90 times oil (engine oil, silicone oil etc.) of its original weight. Besides, the strong PDA chemical bonds enable robust hydrophobicity and adsorption property after 50 cycles of oil adsorption. Liu et al. [62] used the aniline to suit-polymerization onto melamine sponge and grafted with low surface energy dodecyl mercaptan to delivery superhydrophobicity (Figure 8a). After coating, the skeleton became rough while the coated polyaniline (PANI) showed negligible effect on decreasing the pore size (Figure 8c). The obtained superhydrophobic PANI-sponge can adsorb different types of oil or organic solvents about 51–121 times of its initial weight. Moreover, PANI-sponge also was used in oil-in-water emulsion separation. The etching/polymerization can only randomly increase the roughness of the sponge, which may cause the differences in hydrophobicity at different positions on the surface or internal skeleton. However, post-modification on commercial sponge/foam using organic agents makes it difficult to creating uniformity and desired roughness and surface structure. Besides, the mechanical property of the modified sponge/foam is still a concern after several times of oil desorption under compression due to the weak mechanical robustness of the original commercial sponge/foam.

##### Organic–Inorganic Composite Modification Agents

Dispersing inorganic nanomaterials into organic precursors significantly increase the surface roughness and forms the secondary nanostructure after application onto the sponge, which enhances the surface wettability and self-cleaning property [132,143]. Except this, the incorporated inorganic nanomaterials can improve the mechanical performance and the organic agents can bind inorganic nanomaterials onto the skeleton through chemical bonds, resulting in durable hydrophobicity and oil/water separation performance. The widely used inorganic nanomaterials mainly include carbon material (activated carbon [22], graphene oxide (GO) [61,144], graphene (GN) [145,146], carbon nanotubes (CNT) [147], carbon soot (CS) [148], etc.), metal oxide (Al_2_O_3_ [132,149], TiO_2_ [117,150,151], SiO_2_ [122,152], Fe_3_O_4_ [153,154], etc.), and metal organic frameworks (MOFs) [59,155]. Carbon materials have been extensively used in the field of adsorption due to ultra-light, high specific surface area and non-toxicity. Recently, carbon materials have been utilized in 3D-PMs to enhance the adsorption capacity in oily wastewater treatment. Wu et al. [61] prepared the hydrophobic GO-PU sponge with WCA of 133.4° via solution immersion and thermal reduction, which can separate different type oil–water mixtures (bromobenzene, carbon tetrachloride, hexane, etc.) and the separation efficiency is higher than 90%. To increase the oil/water separation efficiency, Li et al. [60] used the γ-methacryloxypropyl trimethoxy silane (KH-570) to modify GO and coated onto PU sponge via dip-coating to prepare the superhydrophobic-superoleophilic sponge. WCA increased from 129° to 161° after modified with KH-570 and the oil absorption capacity was enhanced to more than 39 times of its initial weight, which was improved nearly two times compared to the pure PU sponge. Nguyen et al. [156] used the hydrophobic graphene nanosheets (WCA:132°) and PDMS as cements to increase the bonding force between graphene coating and the commercially sponge via solution immersion method. The WCA of the fabricated sponge was 162°, which was superhydrophobic. The oil or organic solvents adsorption capacity of graphene-based sponge was around 54–165 times its original weight and showed relatively stable solvent adsorption performances after 5 cycles of adsorption-desorption However, the oil adsorption capacity dropped nearly 80% in the second cycle of oil adsorption-desorption (Figure 9a) because the oil adhered to the skeleton, and was difficult to squeeze out via compression. To address this issue, Ge et al. [157] fabricated hydrophobic and conductive GO-wrapped-melamine sponge (MS) via dip coating. This conductive GO-wrapped-MS significantly lower the viscosity of crude oil when the sponge was heated up to 90 °C through applying DC voltage showing fast and large oil adsorption (adsorb 3.87 g crude oil in 6 s) comparing to the original sponge (adsorb 1.5 g crude oil in 8 min) (Figure 9b). Chang et al. [158] mixed the carbon nanotubes (CNTs) with PDMS to modify the PU sponge via dip-coating method obtaining superhydrophobic CNTs-PU sponge and lowered the viscosity of heavy oil under light irradiation which effectively enhanced oil adsorption capacity and self-cleaning performances. 3D-PMs are easily contaminated and adhered by high viscosity oil which is difficult to squeeze out the oil residues and caused surface contamination. Electrically induced joule heating and the light irradiation heating can significantly decrease the viscosity of oils resulting in quick adsorption and surface cleaning, showing high promising in practical oil/water separation. However, the large energy input with relatively low energy utilization, and the thermal stability of the polymer sponge under high temperature are needed to be solved in the future.

Inorganic oxides (SiO_2_, TiO_2_, Fe_3_O_4_, etc.) are sort of important materials in polymer composites due to the high chemical and mechanical stability with relatively low cost [152,159,160]. Thus, they can be utilized to enhance the wettability and mechanical performance of sponges/foams. Ge et al. [161] synthesized hydrophobic SiO_2_ from chlorotrimethylsilane and mixed with polyfluorowax (PFW) to firmly coat onto PU sponge via dip-coating. The incorporation of SiO_2_ nanoparticles made the surface rough and PFW reduced surface energy resulting in superhydrophobicity. Besides, the utilization of PFW fixed SiO_2_ nanoparticles strongly onto PU sponge after curing and delivered robust superhydrophobicity (WCA was only dropped 5° (original: 156°)). The composite sponge showed relatively high mechanical stability by adding certain amount of SiO_2_ nanoparticles after 400 cycles of compression tests under the strain of about 75%. However, chlorotrimethylsilane is inflammable and is easily hydrolyzed into hydrogen chloride, which is unsafe and caused the loss of hydrophobicity and oil/water selectivity. To avoid this, Salehabadi et al. [130] utilized octylsilane to modify SiO_2_ nanoparticles showing that PU sponge reached superhydrophobic state after coating with 2 vol % modified SiO_2_ nanoparticles (Figure 9c_1_). As comparison, the obtained superhydrophobic SiO_2_-coated PU sponge can absorb 61–72 times of its initial weight oils showing high adsorption capacity. However, the un-hydrophobized SiO_2_ tend to agglomerate on the PU sponge at high concentration (3 vol %) (Figure 9c_2_), causing the decrease of wettability and the oil adsorption capacity dropped more than 5 times. Lei et al. [69] firstly deposited Fe_3_O_4_ nanoparticles onto melamine sponge and then coated with lignin by dip-coating to prepare the superhydrophobic composite sponge, which can be driven under magnetic field to quickly separate from oil/water mixture after absorbed the oil (Figure 9d), thereby increasing the separation efficiency. Zeolitic imidazolate frameworks (ZIFs) are a sort of metal–organic framework (MOF) in which porosity and specific surface area are relatively high, showing the potential in adsorption organic compounds [59,155,162]. Kim et al. [155] first carbonized melamine sponge under Ar atmosphere and then the zeolitic imidazolate framework-8 (ZIF-8) was nucleated and grew onto the composite surface in Zn^2+^ and 2-methylimidazole (MIM) solution at room temperature. The obtained ZIF-8 modified sponge showed highly hydrophobic (WCA: 135°) and high specific surface area (BET: 211 m^2^/g) which can absorb 55–136 times oil (pump oil, chloroform etc.) of its initial weight. Inorganic-polymer composite coating increases the hydrophobicity, roughness and the durability of the sponge comparing with only using single inorganic/organic modifications. The incorporated inorganic nanomaterials with unique property can equip the sponge with multifunctional properties such as electrical conductivity, magnetic, photoresponse, which broadens the development of 3D-PMs. 

#### 3.1.2. Preparation of Superhydrophobic 3D-PMs by Self-Assembly

Though the superhydrophobicity can be easily obtained by surface modification with organic/inorganic compounds, the modification layer generates adverse effects on the flexibility and adsorption capacity of the sponge. Moreover, the pore structure (e.g., diameter, porosity) cannot be precisely adjusted through this method [163] and the initial elasticity of the commercial sponge is significantly reduced (Figure 10a,b) [108]. Especially, the incorporated nanomaterials tend to aggregate and cause the generation of cracks at high loading (Figure 10c,d) [108,143]. In order to well-tune the porous structure and improve the mechanical robustness of the sponge, the superhydrophobic sponge can be prepared through assembling of the polymers via freeze-drying, electrospinning, or polymer foaming. 

##### Sponge

Foaming is broadly used to fabricate sponges/foams due to the mass production, ease of fabrication, and abundant pore structures using foaming agents [164,165]. The widely used inorganic foaming agents to fabricate oil/water separation foams are sodium bicarbonate (NaHCO_3_), ammonium carbonate, ammonium nitrite, etc. Among which, NaHCO_3_ is more favorable due to non-toxicity and the formation of uniform and dense pore structures by decomposing into carbon dioxide under thermal heating [166]. Kong et al. [105] utilized polyether polyol, isophorone diisocyanate, and dibutyltin dilaurate as raw materials and NaHCO_3_ as foaming agent to fabricate flexible PU sponge at 100 °C via pyrolysis. Since PU sponge is hydrophilic, by adding a certain amount of hydrophobic γ-(methacryloyloxy)propyltrimethoxysilane (KH-570) modified Al_2_O_3_ hollow spheres during foaming to increase the surface roughness and reduce the surface energy correspondingly, which resulted in highly hydrophobicity (WCA: 144°) (Figure 11b). The obtained composite sponge can absorb 5–37 times oil of its own weight and the oil adsorption capacity is significantly improved (10–50%) compared to PUF without Al_2_O_3_ modification. Besides, Al_2_O_3_-modified PUF showed relatively high oil adsorption stability after 20 times of cycling oil/water separation. However, the fabrication process is relatively complex and requires harsh environmental parameters to foam. Recently, sugar/salt particles and sodium bicarbonate are reported as a pore-forming template to simplify the fabrication of foams/sponge because they can be easily removed through immersing in water/solvent or decomposition under thermal heating without secondary pollution or pore structure damage. Choi et al. [134] used the cube sugar as template to fabricate polydimethylsiloxane (PDMS) sponge with highly hydrophobicity (WCA) (Figure 11c), which can absorb 5 times its initial weight oils (silicone oil, motor oil, transformer oil). The oil (chloroform) adsorption capacity was slightly improved from 9 to 11 times of its original weight while using different size (several hundred micrometers or even centimeters) of sugar particles (granulated + sanding and granulated + black sugar particles) as a template (Figure 11d). Similarly, Zhao et al. [102] used NaCl microparticles as template to prepare the hydrophobic PDMS sponge and h can absorb 5–20 times oil or organic solvents. However, hydrophobic PDMS sponge prevents water from entering and dissolving the sugar/salt microparticles causing sugar or salt residues left into the sponge affecting pore structure, wettability and elasticity. Yu et al. [107] reported that they used the citric acid monohydrate (CAM) as a hard template to create pore structures and fabricate PDMS sponge. CAM can be easily removed by dissolving in ethanol within 6 h because ethanol completely wets the porous PDMS sponge. The PDMS sponge can absorb 7–15 times oil or organic solvents and the separation efficiency reached up to 99.8%. However, the pore size and porosity of the as-prepared 3D-PMs via foaming or templating methods are much larger and smaller than the commercial sponge, correspondingly (Figure 11a) leading to the relatively lower hydrophobicity and oil adsorption capacity. Besides, the pore structure and pore size rely on the size of template with which it is difficult to fabricate uniform and smaller pores. To solve this shortcoming, Wang et al. [167] used the thermally-impacted phase separation method to prepare the low density and high porosity polymer foam by firstly mixing the thermoplastic polyurethane (TPU), dioxane, and deionized water together to form a homogeneous solution and then removing the dioxane and water from the TPU by decreasing the temperature. The superhydrophobic PU foam was successfully prepared via freeze-drying after 24 h. The foam shows highly porosity (91%) and excellent mechanical properties, which only reduced approximately 25% after 1000 compression cycling test at 80% strain. The PU foam can absorb 10–32 times its initial weight in different types of oils or organic solvents. To reduce the cost and achieve eco-friendliness, Wang et al. [52] used biodegradable poly (lactic acid) (PLA) to fabricate the superhydrophobic polymer foam via phase separation and freeze-drying. The result had a large number of mesopores and high surface area (>96.7 m^2^/g by BET test), which could absorb 23 times engine/silicone oil of its original weight.

##### Aerogel

Super-wetting aerogel is reported to treat oily wastewater with the advantages of ultra-light, large specific surface area, large porosity, and relatively large adsorption capacity [127,169,170]. Currently the oil/water separation aerogels included silicon aerogel [171,172,173], carbon aerogel [174,175,176], fiber cellulose aerogel [177,178,179] and composite aerogel [180,181,182] have been reported for oil/water separation because of the unique properties of ultra-light and relatively high specific surface area. However, conventional silicon aerogels such as tetraethoxysilane (TEOS) aerogels are easy to crack under a certain pressure, causing the collapse of the 3D porous structure [183,184]. Gurav et al. [185] used a two-step sol–gel method to fabricate superhydrophobic TEOS-based silica aerogel (WCA:153°) in which firstly dissolved tetraethoxysilane (TEOS) to form sol and added NH_4_F/NH_4_OH to form gel and obtained a 3D structure after a certain period of aging. The superhydrophobicity originated from the utilization of hexamethyldisilazane. Such superhydrophobic aerogel can absorb about 10 times its initial weight in petrol oil. However, the silica aerogel shrinks by nearly 50% and the 3D structure collapsed after compression to squeeze the adsorbed oils (Figure 12a). In order to improve the mechanical properties and maintain its pore structure. Yu et al. [137] used organosilane dimethyldiethoxylsilane (DMDES) and methyltriethoxysilane (MTMS) as co-precursors, which dissolved in ethanol to fabricate the silicon aerogels (density: 0.0897 g/cm^3^) via the sol-gel. The obtained aerogel showed excellent elasticity, which was due to non-polar methyl groups in backbones and the reduction of the inter-chain cohesion (Figure 12b). The abundant hydrophobic methyl groups on MTMS contributes to the superhydrophobicity (WCA 153.6°) of the composite aerogel, thereby the as-prepared aerogel absorbed 6–17 times its own weight in oil or organic solvents and maintained its structure after 10 cycles of oil adsorption and desorption at 60% strain (Figure 12c). Recently, bridged silsesquioxane aerogels are more favorable due to the significantly improved mechanical strength through the formation of covalent bonding between inorganic siloxane and organic functional groups [186]. Chen et al. [187] used terephthalaldehyde (TPAL) and 3-aminopropyl-triethoxysilane (APTES) as a bridging silsesquioxane precursor and MTMS to fabricate the highly hydrophobic silicon composite aerogel (142°) via the catalytic reaction of acetic acid and freeze-drying. The bridged silsesquioxane aerogel can absorb 8–23 times oil or non-polar organic solvents of its initial weight, and are stable after more than 10 times cycling compression at 50% strain. The BET indicates the specific surface area of the bridged silsesquioxane aerogel was 26.7 m^2^/g, which was relatively low, leading to the poor adsorption capacity. To further improve the oil adsorption capacity, varied types of carbon aerogels were fabricated such as graphene, CNTs, cellulose, etc. by hydrothermal reaction [188], chemical reduction [189], and thermal reduction [190]. Ren et al. [191] mixed GO nanosheets with melamine to prepare N-doped graphene aerogel via self-assembly and thermal annealing process (900 °C), and showed highly hydrophobic (WCA: 137.2°). Especially, the specific surface area of graphene aerogel was 852 m^2^/g, and it can absorb 52.6–111.6 times organic solvents of its initial weight. Li et al. [192] used sodium chlorite (SC)-activated poplars catkin (PC) as precursor to prepare PC aerogel, and then carbonized at 1000 °C and successfully obtained the superhydrophobic carbon aerogel with outstanding oil absorption capacity. It can absorb 81–161 times oil (pump oil petroleum ether, etc.) of its initial weight and the adsorption performance remains stable after more than 10 cycles of oil adsorption and desorption. Moreover, PC sponge shows excellent mechanical performance, and can retain its original shape without breaking or collapsing after 500 cycles at 50% strain. Nevertheless, the hydrothermal or carbonization process of preparation aerogels needs relatively high temperatures and the fabrication process of aerogels is always under harsh environment such as supercritical drying, acidic/basic solution, a long period of maintenance, freeze-drying, and relatively high pressure, and produces organic wastewater, which leads to the high cost and may cause secondary pollution. 

To reduce the fabrication cost and minimize the adverse effects on the environment, using nature materials such as cellulose is an alternative way to develop oil/water separation aerogels [54,123,191]. Cellulose originates from natural resources e.g., natural plants, household waste paper, etc., showing the advantages of low cost, high mechanical property, environmentally friendly, biodegradable materials which can be used to fabricate aerogels via freeze-drying [54,193,194]. However, cellulose is hydrophilic and, thus, can be easily wetted by water and needs hydrophobic modification. Mi et al. [64] used bidirectional freeze-drying (control the growth of ice crystals on the y and z axes) and siliconization, obtaining ultra-light (bulk density: 0.0059 g/cm^3^) and superhydrophobic (WCA: 150.3°) cellulose-graphene composite aerogels (Figure 13a). The as-prepared composite aerogel can float on dandelion and shows excellent oil adsorption capacity (80–197 times oil or organic solvents of its own weight) due to multi-layered sheet structure, larger porosity and high BET surface area (47.3 m^2^/g) (Figure 13b). Besides, the composite cellulose aerogel shows high mechanical stability that it recovered to the original size under 50% strain after 100 times cycling compression tests (Figure 13c). However, the irregularity and discontinuity of cellulose leads to the difficulty in precisely controlling the mechanical properties and large scale fabrication [195]. Si et al. [196] used polyacrylonitrile (PAN) nanofibers as the main precursor and crosslinked by the bifunctional benzoxazine (BA-a) to construct elastic networks, and then introduced rigid SiO_2_ nanofibers to enhance its mechanical strength, successfully preparing the nanofibre-assembled cellular aerogels (Figure 13d). The 3D porous structure is relatively stable even after 1000 cycling compressive fatigue test with 60% strain. The nanofibre aerogels can rapidly and effectively separate oil/water emulsions under gravity driven with much higher separation flux comparing with other pressure-driven separation materials due to the numerous multi-level microchannels. By combining with pump, it shows continuous oil/water emulsion separation (Figure 13e,f). Though the aerogels have attracted extensive attentions and the fabrication techniques promote the progress of developing oil/water separation 3D porous materials, its complicated fabrication procedures and the high requirement on temperature and pressure restrict its further application.

### 3.2. Superhydrophilic-Underwater Superoleophobic 3D-PMs

Though superhydrophobic 3D PMs have certain advantages (e.g., large adsorption capacity, high efficiency, recyclable) in oil/water separation from oceans, rivers and lakes, oil easily adheres to the skeleton leading to surface contamination, reduction of hydrophobicity and even loss oil/water separation property [197]. Superhydrophilic-underwater superoleophobic 3D porous materials were developed to overcome oil adhesion by the formed density water film. Oil was prevented to contact with the 3D PMs by the water film thereby keeping the surface clean [198,199]. Moreover, the multi-level interactive three-dimensional porous structures show high efficiency in separating stratified oil/water mixture and oil-in-water emulsion avoiding the secondary pollution which was caused by the utilization of demulsification surfactants. Lv et al. [113] dip-coated different amount of SiO_2_ nanofibers onto melamine formaldehyde (PMF) sponge to adjust the pore size, resulting in the formation of hierarchical pore structures (such as major pores: 100~200 µm, nanoscrolls: 30 nm~2 µm). The obtained sponge with hierarchical pore structure shows highly efficiency (the oil rejection rate is around 99.46%) on demulsification of *n*-hexane emulsions. Superhydrophilicity-underwater superoleophobicity (OCA: 162.57°) of PMF composite sponge are tuned via the in-situ growth of hydroxide by depositing polydopamine (PDA) and polyethyleneimine (PEI). Water droplets quickly wet the sponge within 25 ms while the underwater oil contact angle (1,2-dichloroethane) is measured of 162.57°. The superhydrophilicity-underwater superoleophobicity enables the quick separation of light oil from water surface and oil from oil-in-water emulsions under gravity driven. However, the double-layered hydroxide (LDH) is positively charged which is not suitable for the separation of negatively charged emulsion due to the neutralization reaction between positive surfaces and negative liquid causing oil contamination and the reduction of flux, and the preparation process is complicated. To solve this problem, Wang et al. [89] mixed cellulose powder with pore-forming agent (Na_2_SO_4_·10H_2_O) and then regenerated the fibers to fabricate superhydrophilic-underwater superoleophobic cellulose sponge via foaming and freeze drying. The cellulose sponge has excellent demulsification performances by gravity driven (Figure 14a) and maintains high-flux (91 L m^−2^ h^−1^) and separation efficiency (99.94%) after more than six cycles. The FTIR (Figure 14b_4_) and UV-VIS analyses (Figure 14b_5_) illustrate that the oil was removed completely after the emulsion passing through the sponge (Figure 14b). Due to the strong hydrogen bonding affinity between water molecules and cellulose sponges, oil is replaced by the strong competitive adsorbed water showing excellent self-cleaning and anti-pollution ability when the oil-contaminated sponge was immersed into water (Figure 14c).

### 3.3. Superhydrophilic-Superoleophobic 3D-PMs

In oil mining industry, water droplets disperse into oil and water-in-oil emulsions cause the impure of oil. Thus, removing water from the bulk oil is critical for oil purification. The superhydrophobic and superhydrophilic-underwater superoleophobic 3D-PMs are oleophilic which cannot remove minor water from oil/water mixture or water-in-oil emulsion. A new type of superhydrophilic-superoleophobic 3D-PMs have been developed by changing the structure of the super-wetting coating. The mechanisms of designing this unique surface are widely investigated. Okada et al. [200] uniformly coats the fluoroalkylated acrylic oligomer (FAAO) on the resin substrate and successfully prepared the surface with hydrophilic and oil-repellent properties. When water contacts the composite surface, the hydrophilic unit (–COOH) of FAAO is attracted by water molecules causing surface wetting. On the contrary, the hydrophilic unit is flattened by contacting with oils while the fluoroalkyl unit exhibits erect state showing oil repellency. Brown et al. [201] designed a new type flip-flop coating by the layer-by-layer deposition and each layer was interacted by the strong electrostatic interaction. Cationic polydiallyldimethylammonium chloride (PDDA) is chosen as the base layer to bind strongly onto negatively charged glass surface and SiO_2_ nanoparticles via electrostatic interactions. Then the second (negatively charged SiO_2_ nanoparticles) and third layers (cationic PDDA) are assembled onto the surface strongly by the electrostatic force. Finally, the fluorosilane were deposited on the last layer to deliver hydrophilic-oleophobicity (WCA < 5°, OCA = 157°). The mechanism of delivering such unique wettability can be explained as follow that the –COOH group of perfluorooctanoic acid is more polar than CF_x_ which preferentially adheres to the surface of the substrate while the fluorinated tail heading out to form low surface energy interface showing oil repellency [201]. Pan et al. [202] utilized sodium atoms (Na) to replace the hydrogen atoms in –COOH group thereby molecules with low surface energy tails and high surface energy heads were formed resulting in superhydrophilicity and oil repellency (Figure 15a).

Following the above mechanisms, a series of superhydrophilic-superoleophobic sponge/foams have been developed for oily wastewater treatment. Xu et al. [115] dip-coated heptafluoroantimonic acid modified TiO_2_ sol and SiO_2_ NPs onto PU sponge to increase surface roughness. After that, the composite polyurethane (PU) sponge was treated by ammonia vapor to transfer superhydrophilicity-superoleophobicity to superhydrophobic-superoleophobicity. The modified PU sponge can easily remove water from the bottom of the diesel oil without any oil residues left in water (Figure 15b). Su et al. [116] fabricated superhydrophilic-superoleophobic melamine sponge (Figure 15c,d) by using sodium perfluorononanoate (PFNA) and Fe_3_O_4_ nanoparticels via dip-coating approach obtaining highly oleophobicity (OCA: 144°).The obtained sponge can eliminate water from oil/water mixtures by gravity driven with the separation efficiency reach to 97.6% (pump oil) and 94% (pump oil). This method does not require pretreatment (wet by water) or the formation of oil repellent water films. Moreover, combining with the peristaltic pump, the superhydrophilic-superoleophobic sponge can continuously remove large amounts of water pollutants from bulk oil by (Figure 15e). 

### 3.4. 3D PMs with Switschable Super-Wettabilityunder External Stimulation

Either superhydrophobic or superhydrophilic 3D PMs, they only have one single wetting property and can only absorb or separate specific liquids. In addition, oils, especially the high viscos oil (e.g., crude oil) are easily adhering onto the superhydrophobic surface causing the reduction of adsorption capacity and difficult to desorb [203]. Smart 3D porous materials with switchable wettability are receiving more and more attention due to the advantages of responding to external stimulation, ease of adsorb-desorption, self-cleaning, multi-functional oil/water separation under varied environment. A series of 3D-PMs have been reported to utilize in oil/water separation which can switch their wettability under the stimulation of light [204], heat [205], pH [206] showing ease of desorption after oil absorption and self-cleaning (Table 3). Li et al. [204] reported to fabricate light responding PU sponge through immersing into octadecanoic acid and titanium dioxide (TiO_2_) solution under ultrasonication and obtained coral-like cluster rough structures with superhydrophobicity (WCA: 151°). The superhydrophobic composite sponge (151.1°) was switched into superhydrophilic (0°) under UV irradiation by from the generation of Ti^3+^, oxygen vacancy and the formation of hydrogen bonds. This smart TiO_2_ PU sponge can absorb 27–60 times oil or organic solvents of its original weight and shows certain antifouling and self-cleaning properties. The adhered oil on the sponge skeleton causing the decreasing of separation efficiency which can be released under UV irradiation attributing to the conversion of hydrophobicity to hydrophilicity. Kim et al. [117] further investigated the mechanisms of oil desorption under UV irradiation indicating that the oil desorption was attributed to the growth and diffusion of the generated bubbles from the dissolved oxygen and nitrogen under the catalysis of TiO_2_. The bubbles wrapped oil to increase its diffusion power and thus desorb from porous materials surface into water (Figure 16a,b).

Comparing with UV irradiation which requires the generation of UV light, thermal and pH response can also be utilized to tune the super-wetting property of surfaces relatively simply. Poly (*N*-isopropyl acrylamide) (PNIPAAm) is a type of thermal response material with relatively low hydrophilic and hydrophobic transition temperature (LCST) of 32 °C [205]. When the temperature is less than the LCST, PNIPAAm forms hydrogen bonds with water molecule to hydrate, swell and outstretch resulting in hydrophilicity; when the temperature is higher than the LCST, the C=O group combines with the N-H group to form intramolecular hydrogen bond resulting in hydrophobicity [205]. Lei et al. [205] grafted hydrophobic octadecyltrichlorosilane (OTS) modified PNIPAAm onto melamine sponge to artificial control wettability by changing temperature. Water contact angle increased from 0° to 150° when the temperature rises from 25 to 40 °C showing superhydrophobicity. Oil can be quickly adsorbed at 37 °C (the modified sponge can absorb about 35 times pump oil/peanut oil/gasoline of its initial weight) while the oil can be quickly and completely discharged after applying a slightly pressure at 20 °C (Figure 16c_1_). Furthermore, the thermal response performance was stable after 5 adsorption/desorption cycles (Figure 16c_2_). pH responded and switchable super-wetting melamine sponge was developed by grafting poly-4-vinylpyridine (PVP) [206]. Water contact angle gradually rises from 0° to 137° when pH increases from 1 to 7. When pH = 7, WCA reached to 137°, achieving a transition between superhydrophilicity and superhydrophobicity. PVP deprotonated and agglomerated at pH=7 showing hydrophobicity and it can quickly absorb oil stains; the PVP protonated pyridyl group stretches at pH=1 showing hydrophilicity and rapidly releasing of oil stains. By changing the environmental parameters, the sponge with switchable wettability provides a good solution for the treatment of highly viscous oil e.g., crude oil. 

## 4. Conclusions and Perspective

In this article, we briefly reviewed the typical materials and techniques which are utilized in the recent development of superhydrophobic-superoleophilic, superhydrophilic-superoleophobic, superhydrophilic-underwater superoleophobic, switchable super-wetting polymeric oil/water separation three dimensional (3D) porous materials such as sponge, foam and aerogels. By reviewing the effects of wettability on oil/separation, the desired wettability on dealing the specific oil/water mixtures was developed correspondingly. There are a variety of polymers can be used to fabricate super-wettable 3D-PMs through similar techniques. However, the requirements of wettability are not the same due to their unique roles in separating specific oil/water mixtures. Surface wettability be tuned through the low surface energy polymers and nanomaterials. The utilized nanomaterials significantly enhanced the roughness to improve the wettability and separation efficiency. Moreover, the incorporated nanomaterials can also be applied to improve the mechanical stability of 3D-PMs. The organic–inorganic nanomaterials modification coatings show great potential in developing high performance 3D-PMs to meet the advanced requirements in current and the future oily wastewater treatment.

However, there are some challenges in developing polymeric 3D-PMs and their utilization in oil/water separation in the following aspects: (i) materials and technologies that are suitable for large-scale, simple and low-cost production, environmentally friendly, and avoid secondary pollution; (ii) the actual oily wastewater is complicated, especially with high adhesion and high density oils, one of the significant shortcomings that restricts the application in oil/water separation is oil contamination, thus developing superwetting polymeric 3D-PMs with anti-oil-pollution is critical; (iii) squeezing the adsorbed liquid after long-term filtration or adsorption is a fundamental requirement for cycling oil/water separation, thus, developing 3D-PMs with excellent mechanical stability and elasticity is crucial; (iv) the stability of superwetting coating determines the efficient selectivity of 3D-PMs, but they are easily peeled off from the surface, leading to the loss of super-wettability. We expect that, in the near future, the above issues can be solved gradually to promote the applications of super-wetting 3D-PMs in oily wastewater treatment with high efficiency, anti-fouling, and mechanical robust performance.

## Figures and Tables

**Figure 1 polymers-11-00806-f001:**
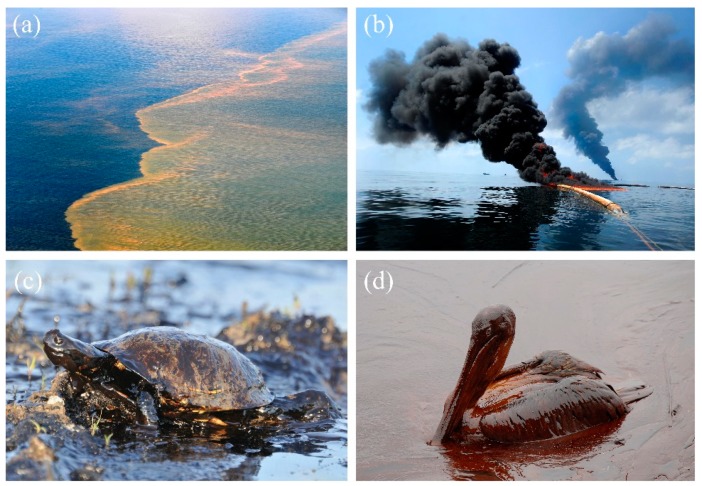
Gulf of Mexico oil spill event [7]: (**a**) marine ecosystems are polluted [10], (**b**) preventing the rapid spread of oil via oil fences and burning, and wide range of smoke causing serious air pollution [13]; reproduced with permission from the public domain in the United States. (**c** and **d**) the turtle [11] and brown pelican [12] contaminated with leaked crude oil on the sea; reproduced with permission from the Thinkstock and the Creative Commons Attribution-Share Alike 2.0 Generic.

**Figure 2 polymers-11-00806-f002:**
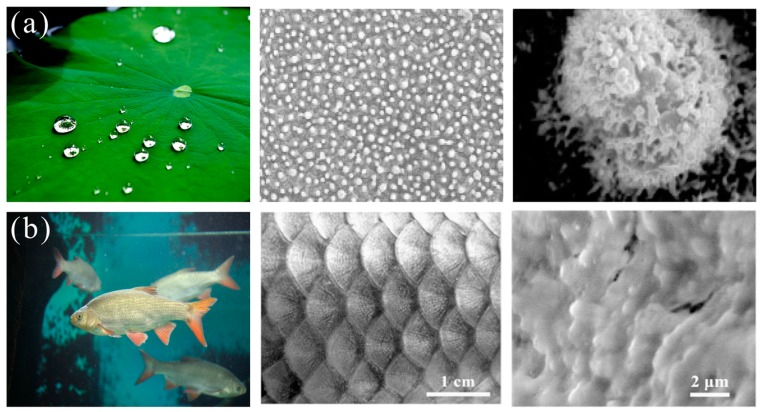
Biological surfaces with super-wettability and their structures: (**a**) lotus leaf [29]; reproduced with permission from Wiley. (**b**) fish scales [30]; reproduced with permission from Wiley.

**Figure 3 polymers-11-00806-f003:**
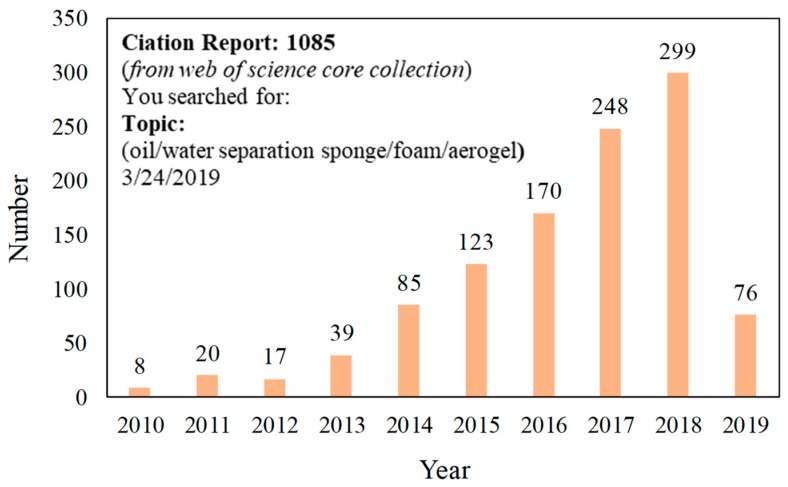
The published articles from 2010 to February 2019 indexed in ISI Web of Science by searching “oil/water separation sponge/foam/aerogels”.

**Figure 4 polymers-11-00806-f004:**
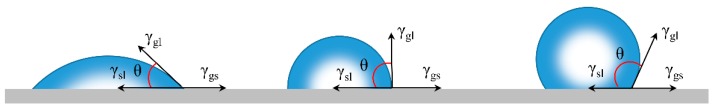
Schematic illustration of Young’s model [73]; the contact angles (*θ*) formed by liquid droplets on a smooth homogeneous solid surface in air (left). A hydrophilic (middle) surface with *θ* < 90° and a hydrophobic surface with the *θ* > 90°(right); reproduced with permission from Springer.

**Figure 5 polymers-11-00806-f005:**
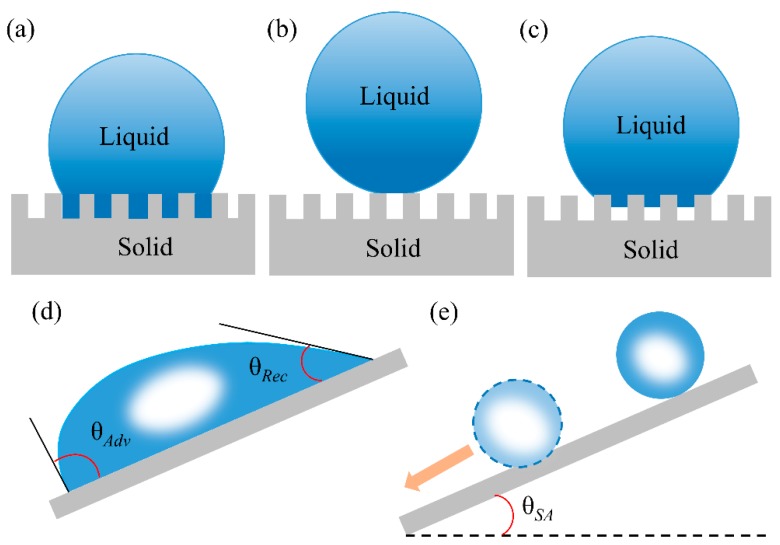
Effect of solid surface structure on wetting behavior: (**a**) Wenzel mode [70], (**b**) Cassie–Baxter mode [76], (**c**) intermediate states between the Wenzel and the Cassie modes [33]; reproduced with permission from Wiley. (**d**) Illustration of the “tilted plate” method to measure advancing/receding angle, respectively when the drop just starts to move [73]; reproduced with permission from Springer. (**e**) Illustration of the sliding angle [83]; reproduced with permission from the Royal Society of Chemistry.

**Figure 6 polymers-11-00806-f006:**
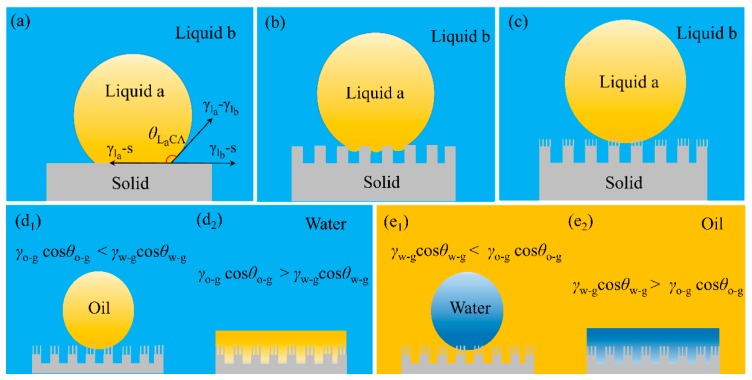
(**a**–**c**) Illustration of the behavior wetting effect by surface structure of the solid substrates in oil/water [30]. (**a**) smooth surface, (**b**) microstructure surface, (**c**) micro/nanohierarchical surface in liquid b; reproduced with permission from Wiley. (**d**,**e**) Illustration of the theoretical extreme wettability of solid substrate surface in water/oil [87]. The substrate exposed in the water or oil shows four oil–water states: (**d_1_**) superoleophobic, (**d_2_**) superoleophilic, (**e_1_**) superhydrophobic, (**e_2_**) superhydrophilic. o-g: oil and gas interface, w-g: water and gas interface; reproduced with permission from American Chemical Society.

**Figure 7 polymers-11-00806-f007:**
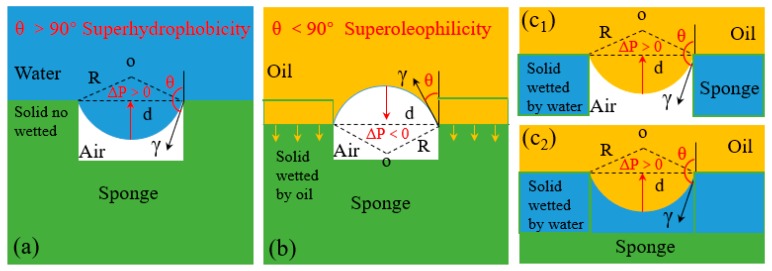
Schematic illustration of liquid-wetting modes of 3D materials [74,89,90]; (**a**) because Δ*p* is positive, the superhydrophobic 3D material repels the water penetration by the trapped air. (**b**) because Δ*p* is negative, the oil can permeate the superhydrophobic 3D material spontaneously. (**c**) there are two states of the oil supported by superhydrophilic 3D materials: (**c_1_**) the oil layer supported by the trapped air in cavities, (**c_2_**) the oil layer supported by the trapped water in cavities; reproduced with permission from the Royal Society of Chemistry.

**Figure 8 polymers-11-00806-f008:**
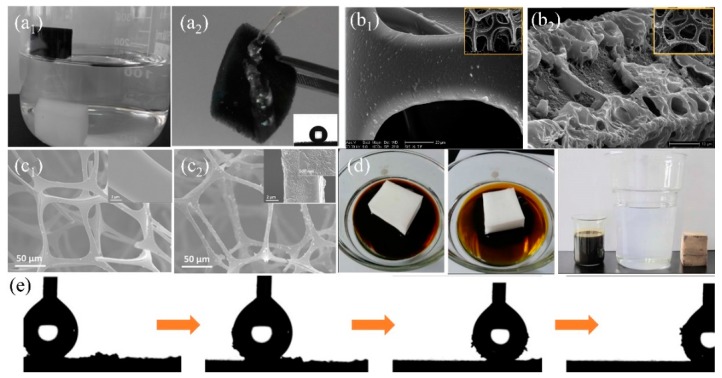
(**a**) The photograph of (**a_1_**) original melamine sponge (sinking under the water) and PANI-melamine sponge (floating on the water), (**a_2_**) the surface of modified sponge rebounding water (the insert is water contact angle (WCA) image) [62]; reproduced with permission from Wiley. (**b**) SEM images of (**b_1_**) the original polyurethane (PU) sponge and (**b_2_**) the PU sponges modified with methyltrichlorosilane [108]; reproduced with permission from Royal Society of Chemistry. (**c**) SEM images of (**c_1_**) original sponge and (**c_2_**) the PANI-melamine sponge [62]. (**d**) Photograph of the KH-570 PU foam absorbs the crude oil [140]. (**e**) Optical images of the water droplet (5 µL) removing carbon black particles from the surface of FAS-PU sponge [124]; reproduced with permission from Wiley.

**Figure 9 polymers-11-00806-f009:**
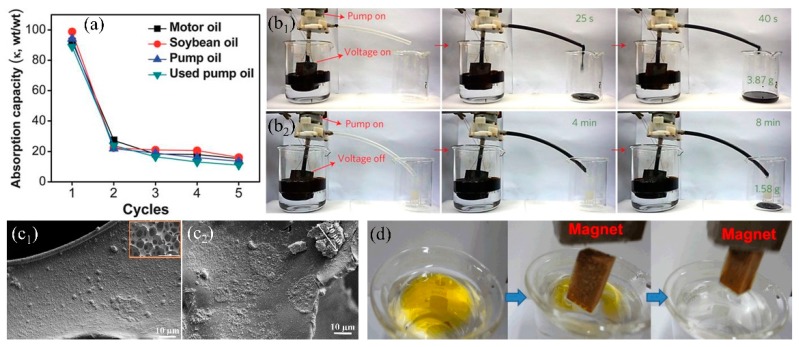
(**a**) Absorption recyclability of the sponge modified with graphene for oils [156]. (**b**) Comparison of efficiency of the viscous crude oil continuous collection device: (**b_1_**) applied voltages of 17 V, (**b_2_**) applied voltages of 0 V [157]. (**c**) SEM images of the PU sponge modified by hydrophobic SiO_2_, (**c_1_**) and un-hydrophobic SiO_2_ (**c_2_**) [130]; reproduced with permission from Springer. (**d**) Photograph of the Fe_3_O_4_ melamine sponge absorbs the oil (yellow) by magnetic driven [69]; reproduced with permission from Elsevier.

**Figure 10 polymers-11-00806-f010:**
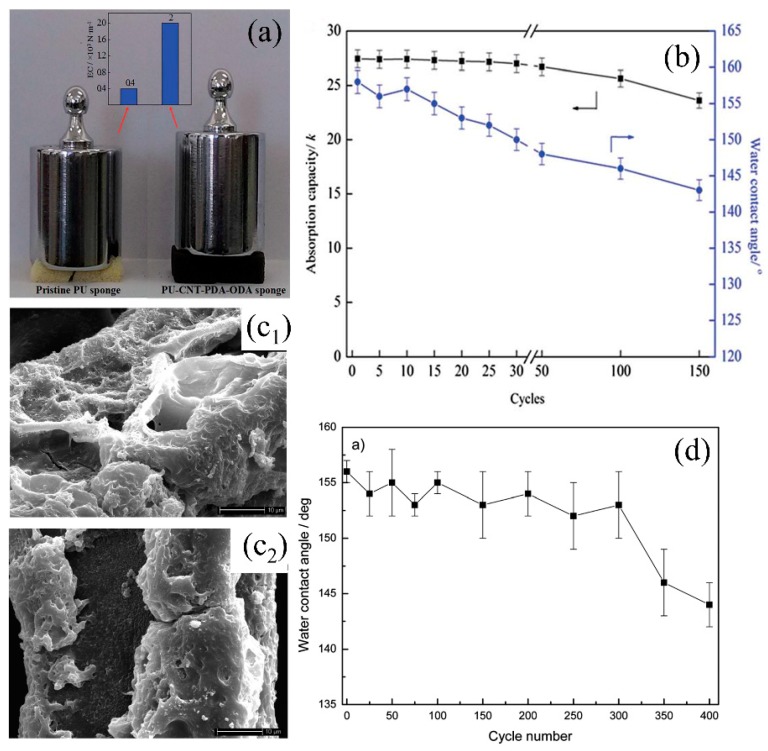
(**a**) Photograph of the elasticity comparison of original sponge and CNT–PDA–ODA PU sponge, (**b**) variation of the CNT–PDA–ODA PU sponge’s WCA and absorption capacity with cycles [143]. (**c**) SEM images of the methyltrichlorosilane-PU sponge for 300 (**c_1_**) and 400 (**c_2_**) cycles after water/oil separation, (**d**) variation of the methyltrichlorosilane-PU sponge’s WCA with cycles [108]; reproduced with permission from Royal Society of Chemistry.

**Figure 11 polymers-11-00806-f011:**
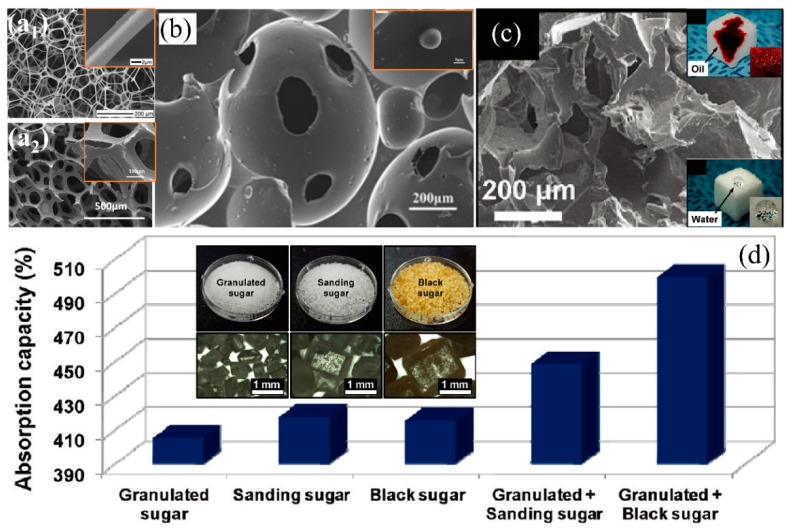
(**a**) SEM images of (**a_1_**) the melamine sponge [168] and (**a_2_**) polyurethane sponge [132]; reproduced with permission from American Chemical Society and Springer. (**b**) SEM images of the pure PUF sponge (the insert is the Al_2_O_3_/PUF sponge) [105]; reproduced with permission from Elsevier. (**c**) SEM images of the PDMS sponge (the inserts are photograph of the sponge hydrophobicity and oleophilicity), (**d**) the effects of sugar templates on absorption capacity of PDMS sponges with various (the insert is images of the sugar particles with different size) [134]; reproduced with permission from American Chemical Society.

**Figure 12 polymers-11-00806-f012:**
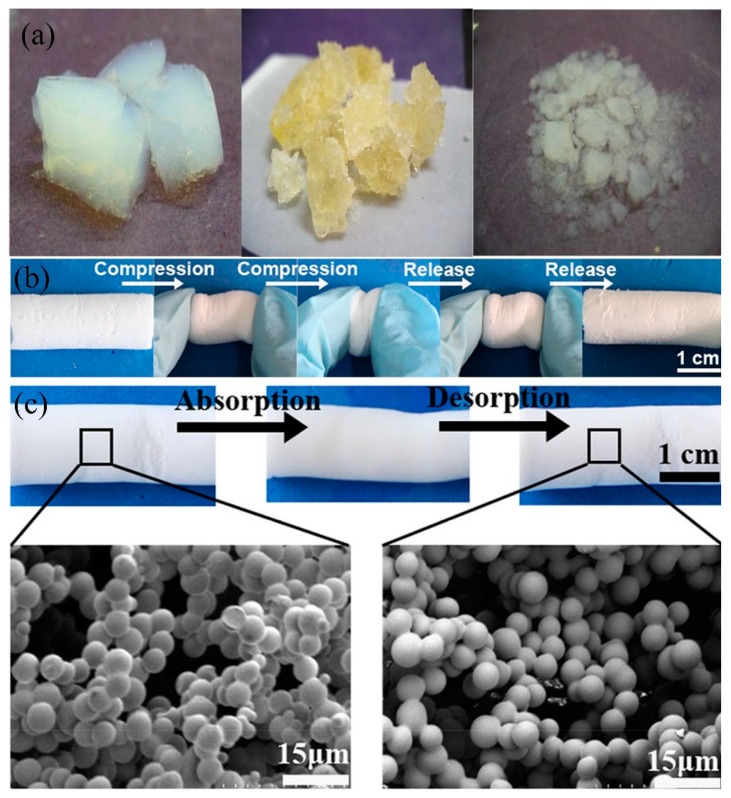
(**a**) Photographs of the various stages of tetraethoxysilane (TEOS) aerogel, before absorption (left), during desorption (middle and right) [185]. (**b**) Photograph of the compression/recovery process of the dimethyldiethoxylsilane–methyltriethoxysilane (DMDES–MTMS) aerogel. (**c**) SEM images of the adsorption (before) and the desorption (after) of aerogel [137]; reproduced with permission from Springer.

**Figure 13 polymers-11-00806-f013:**
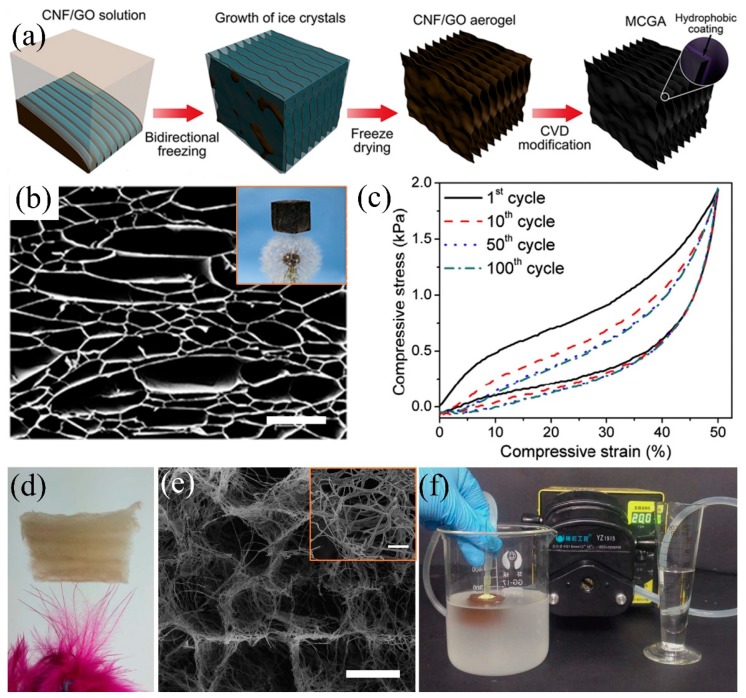
(**a**) Schematic diagram of the bidirectional freeze-drying process to prepare the cellulose-graphene composite aerogel, (**b**) SEM image of the cellulose-graphene aerogel (the insert is the photograph of aerogel on the dandelion), (**c**) elastic cycle performance of aerogels by 100 cycles [64]; reproduced with permission from Elsevier. (**d**) Photograph of the cellular aerogel on the feather, (**e**) SEM image of the cellular aerogels, (**f**) illustration of continuous oil-in-water emulsion separation by cellular aerogel [196]; reproduced with permission from Nature.

**Figure 14 polymers-11-00806-f014:**
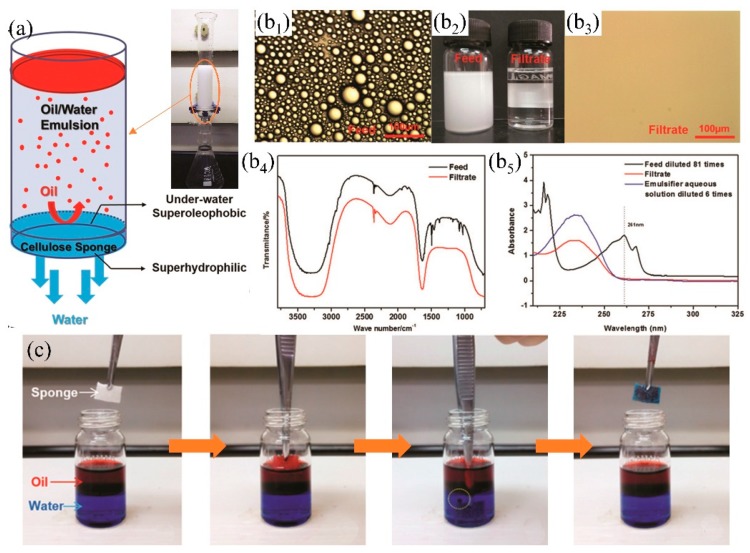
(**a**) Schematic diagram of the cellulose sponge separates oil/water (toluene-in water emulsion), (**b**) the emulsion separation result: (**b_2_**) comparison of the emulsion before (left) and after (right) filtration, photograph (**b_1_**) and (**b_3_**), FTIR (**b_4_**) and UV-VIS (**b_5_**) of emulsion before and after filtration, (**c**) self-cleaning and resistance to oil pollution of cellulose sponge [89]. reproduced with permission from Royal Society of Chemistry.

**Figure 15 polymers-11-00806-f015:**
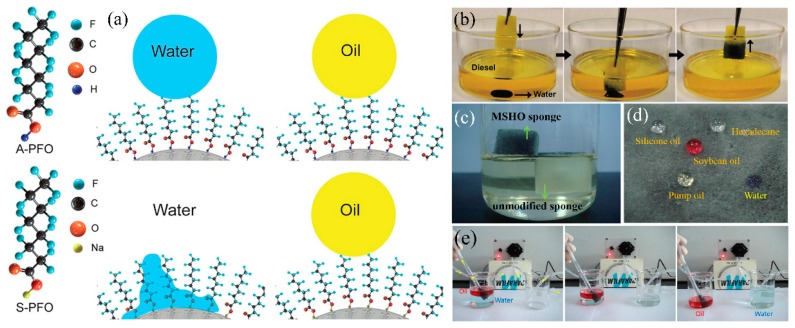
(**a**) Schematic diagram of superhydrophobic/superoleophobic conversion to superhydrophilic/superoleophobic [202]; reproduced with permission from Royal Society of Chemistry. (**b**) Photograph of the superhydrophilic/oleophobic sponge removes the water from oil [115]; reproduced with permission from Wiley. (**c**) Photograph of the original sponge (sinking under the oil) and magnetic superhydrophilic/oleophobic (MSHO) sponge (floating on the oil), (**d**) photograph of water and different oil placed on the MSHO sponge, (**e**) photograph of the MSHO sponge with pump to quickly separate oil/water [116]; reproduced with permission from Elsevier.

**Figure 16 polymers-11-00806-f016:**
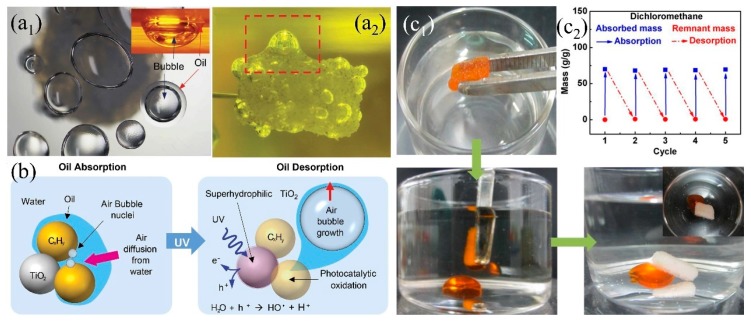
(**a**) Photograph of (**a_1_**) top view of desorbed oil/bubbles (the inset shows the side view), (**a_2_**) oil/bubble desorption (about 24 h), (**b**) schematic diagram of oil desorption under UV light response [117]; reproduced with permission from Nature. (**c_1_**) Photograph of rapid desorption of oil by slightly squeezing the sponge at 20 °C, (**c_2_**) cycling performance of OTS/PNIPAAm sponge adsorption/desorption at different temperatures [205], reproduced with permission from American Chemical Society.

**Table 1 polymers-11-00806-t001:** The techniques reported to develop different super-wetting 3D porous materials and the treated specific type of oil/water mixtures.

Super-wetting Types	3D Porous Structure	Techniques	Types of Oil/Water Mixture	Reference
Superhydrophobic superoleophilic	Sponge Foam Aerogel	dip-coating self-assembly chemical bonding vapor deposition polymerization foaming freeze-drying sol−gel process phase separation	layered oil–water mixtures oil-in-water emulsion	[53,62,67,99,100,101,102,103,104,105,106,107,108,109,110,111,112]
Superhydrophilic underwater superoleophobic	Sponge Foam Aerogel	dip-coating freeze-drying hydrothermal treatment	layered oil–water mixtures oil-in-water emulsion	[89,113,114]
superhydrophilic-superoleophobic,	Sponge Foam	dip-coating self-assembly	layered oil–water mixtures water-in-oil emulsion	[115,116]
switchable super-wettable	Sponge Foam	dip-coating self-assembly	layered oil–water mixtures	[117,118,119,120,121]

**Table 2 polymers-11-00806-t002:** Summary and comparison of typical examples of the superhydrophobic-superoleophilic oil/water separation 3D-PMs.

	Polymeric 3D-PMs	Preparation Methods	WCA [°]	Absorbates	Absorption Capacity [g·g^−1^]	Reference
1	PDMS-PU sponge	Solute on immersion	>150	hexane, toluene, octadecene, silicone oil, motor oil	45–70	[125]
2	PANI/n-dodecylthiol coated melamine sponge	situ-polymerization	≈152.3	pump oil, vegetable oil, petroleum ether, chloroform, n-hexane, etc.	51–122	[62]
3	Poly (furfuryl alcohol) -melamine sponge (MS)	soaking-polymerization	138–145	chloroform, toluene, CCl_4_, n-hexane, methylsilicon oil, paraffin oil, etc.	75–160	[68]
4	LDH/PDA/Fe_3_O_4_/OM -PU sponge	self-polymerization dip-coating method	158	pump oil, toluene, lubricating oil, olive oil, rapeseed oil, castor oil, etc.	34.2–53.6	[129]
5	THF/SiO_2_-PU sponge	ultrasonic-assisted dip coating (UADC)	155	motor oil, kerosene, hexane and castor oil.	51–72	[130]
6	KH-570/GN-PU sponge	dip-coating method	161	soybean, diesel, and pumping oils.	39	[60]
7	Trimethoxysilane/GO-PU sponge	solvothermal treatment	160	Lubrication oil, n-hexane, crude oil, diesel oil.	25.8–44.1	[131]
8	TMC/Al_2_O_3_/PEI/palmitic acid -PU sponge	3-step modification progress	161	Soybean oil, diesel oil, n-hexane, compressor oil, dichloromethane, etc.	16.5–29.9	[132]
9	Fe_3_O_4_/Actyflon-G502-PU sponge	ultrasonic-assisted dip coating (UADC)	153	Hexane, isooctane, toluene, dichloromethane, etc.	25–87	[133]
10	oleic acid/TFAA/TiO_2_ -PU sponge	dip-coating method	161.1	Methanol, ethanol, hexane, DMSO, DMF, Acetone, chloroform, THF, pump oil, etc.	37.2–88.1	[117]
11	Polydimethylsiloxane (PDMS) Sponge	Sugar template method	>120–130	Chloroform, hexane, motor oil, silicone oil, Toluene, transformer oil, methanol, etc.	4–11	[134]
12	Nanocellulose sponge (treatment with stearoyl chloride)	freeze-drying	160	Dichloromethane, silicone oil, toluene, ethanol, acetone, n-hexane, n-octane, etc.	25–55	[135]
13	MCC and MC Silica sponge	sol−gel process	>160	Dichloromethane, n-hexane, gasoline, Petroleum ether, methylbenzene, diesel oil.	3–14	[136]
14	MTES and DMDES aerogels	sol–gel reaction	153.6	Hexane, ethanol, methanol, soybean oil, Chloroform, dichloromethane, etc.	6.83–16.93	[137]
15	Ultralight electrospun cellulose sponge	Electrospinning and freeze-drying	141.2	DMSO, DMF, toluene, chloroform, DMC, acetone, methanol, hexane, pump oil, etc.	15–37	[128]
16	Bacterial Cellulose Aerogels	freeze-drying	146.5	Chloroform, plant oil, dichloromethane, n-hexane, gasoline, paraffin oil, etc.	80–185	[127]
17	cellulose/graphene aerogel	bidirectional freeze-drying	>150	Chloroform, benzene, dichloromethane, DMSO, olive oil, gasoline, octane, etc.	80–197	[64]
18	hydrophobic Al_2_O_3_ polyurethane foam sponge	foaming technology	144	Chloroform, methylbenzene, bean oil, diesel oil, tetrachloromethane, etc.	6.8–37	[105]
19	SiO_2_/PVA/PDMS	electrospinning freeze-drying	>156	hexane, chloroform, octane, toluene	45–91	[138]
20	polyethylene (HDPE)/ microfiber bundles (PMBs)	phase separation centrifugation	141	Pump oil, silicone oil, methanol, etc.	3.34–7	[139]

**Table 3 polymers-11-00806-t003:** Summary and comparison of typical examples of the polymeric three dimensional (3D) porous materials with switchable super-wettability.

	Polymeric 3D-PMs	Preparation Method	Response	WCA [°]	Absorbates	Absorption Capacity [g·g^−1^]	Reference.
1	TiO_2_/octadecanoic acid -PU sponge	ultrasonic-assisted dip coating (UADC)	UV-light temperature	0–151	Hexane, toluene, isooctane, etc.	27–60	[204]
2	PNIPAAm-melamine sponge	solution immersion	Temperature 25–40	0–150	Gasoline, peanut oil, dichloromethane Petroleum ether, acetone etc.	30–70	[205]
3	poly(2-vinylpyridine)/PDMS -PU sponge	block copolymer-grafting strategy	pH 2–6.5	0–150	–	–	[120]
4	poly (4-vinylpyridine) -melamine sponge	atom transfer radical polymerization	pH 1–7	0–135	–	–	[206]
5	Pyridine polymers-PDMS	solution curing polymerization	pH 2–7	10–138	acetone, diesel, methanol, petroleum ether, etc.	6.2–43	[119]
6	HS(CH_2_)_11_CH_3_/HS(CH_2_)_10_COOH/HS(CH_2_)_11_OH/Ag-PU sponge	self-assembled monolayers	pH 1–14	0–150	n-hexane, tetrahydrofuran, trichloromethane, petroleum ether etc.	21.5–81	[121]

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
