# Peer review of "Superwetting Polymeric Three Dimensional (3D) Porous Materials for Oil/Water Separation: A Review"

_polymers, 2019, doi:10.3390/polym11050806_

Round 1
Reviewer 1 Report
To the best of my knowledge, there are several reviews focus on “Porous Materials for Oil/Water Separation”. This review attempts to give an overview of superwetting polymeric 3D porous materials. Polymer porous materials is a promising and widely used oil/water separation materials, and also widely used for other application, such as stain sensors. The present authors attempted to squeeze in several topics into this manuscript with only 139 references. The text - in its current shape - fails to guide the reader through the problem and fails to give the fabricate of the governing problems in porous materials for oil/water separation materials:
1. Overall it should gave a broad scope on oil/water separation materials as well as is significates. Furthermore, there are no explanations about the advantage of polymer 3D porous materials and the state of the art. Moreover, as a review, I found that there are few recent references. The relevant references should be included. Some important references about polymer 3D porous materials and oil/water separation must be cited, for example, ACS Appl. Mater. Interfaces, 2019, DOI: 10.1021/acsami.9b02285; ACS Sustainable Chem. Eng., 2018, 6, 12580; Macromolecular Rapid Communications 2018, 39, 1800635; ACS Sustainable Chem. Eng., 2018, 6, 9866; Applied Materials Today 2017, 9, 77 and so on.
2. There is no Keywords. More appropriate keywords should be given.
3. The fabrication method for the polymer 3D porous materials should be reviewed. For example, research about phase separation, a facile and template-free method to form polymer foams with high porosity, should be mentioned in the article.
4. Page 2, line 39, “low coast” should be “low cost”. Please careful check the manuscript.
5. BET is a method to test the specific surface area and pore distribution of the 3D-polymer material which should be mentioned in this article.
6. The accuracy for some text cited from the original publication should be checked as well as for the description in some places should be checked (For example, the authors menationed that "Wang et al.[103] firstly wrapped CNTs with..."). The text gives details of several references, but does not analyses similarities and/or differences between the influencing parameters.
7. The English-use should be improved in some places.
Author Response
Reviewer 1
Comments and Suggestions for Authors
To the best of my knowledge, there are several reviews focus on “Porous Materials for Oil/Water Separation”. This review attempts to give an overview of superwetting polymeric 3D porous materials. Polymer porous materials is a promising and widely used oil/water separation materials, and also widely used for other application, such as stain sensors. The present authors attempted to squeeze in several topics into this manuscript with only 139 references. The text - in its current shape - fails to guide the reader through the problem and fails to give the fabricate of the governing problems in porous materials for oil/water separation materials:
1. Overall it should gave a broad scope on oil/water separation materials as well as is significates. Furthermore, there are no explanations about the advantage of polymer 3D porous materials and the state of the art. Moreover, as a review, I found that there are few recent references. The relevant references should be included. Some important references about polymer 3D porous materials and oil/water separation must be cited, for example, ACS Appl. Mater. Interfaces, 2019, DOI: 10.1021/acsami.9b02285; ACS Sustainable Chem. Eng., 2018, 6, 12580; Macromolecular Rapid Communications 2018, 39, 1800635; ACS Sustainable Chem. Eng., 2018, 6, 9866; Applied Materials Today 2017, 9, 77 and so on.
Response: We thank the reviewer’s comments. We have added the information on page 2 and highlighted in blue.
2. There is no Keywords. More appropriate keywords should be given.
Response: The Keywords have been added on page 1 and highlighted in blue.
3. The fabrication method for the polymer 3D porous materials should be reviewed. For example, research about phase separation, a facile and template-free method to form polymer foams with high porosity, should be mentioned in the article.
Response: Thank you for your comments. The fabrication methods have been added on page 10 and highlighted in blue.
4. Page 2, line 39, “low coast” should be “low cost”. Please careful check the manuscript.
Response: Thank you for pointing out this for us. We have corrected the wrong word. Also, we have gone through the whole manuscript to correct the wrong words and highlighted the corrections in yellow.
5. BET is a method to test the specific surface area and pore distribution of the 3D-polymer material which should be mentioned in this article.
Response: Thank you for comments. The characterization of specific surface area via BET has been added in the manuscript and highlighted in blue.
6. The accuracy for some text cited from the original publication should be checked as well as for the description in some places should be checked (For example, the authors menationed that "Wang et al.[103] firstly wrapped CNTs with..."). The text gives details of several references, but does not analyses similarities and/or differences between the influencing parameters.
Response: Thank you for your comments. We have re-organized our references to accurately corresponding to the description in the main text. We analyzed the references we used in the manuscript to illustrate their differences or similarities.
7. The English-use should be improved in some places.
Response: Thank you for your comments. We have polished the language of the whole manuscript and the changes have been marked in yellow.

Reviewer 2 Report
Do we need another review of oil-water separation devices? I feel that there are already enough.
Generally the written English is understandable but there are many mistakes in the text – too many for the reviewer to go through the text and highlight them. I recommend either taking more time to ensure that the English is as good as possible, or using an English editing service.
Photos shown in the figure 1 are very nice and add to the review article. However, I don’t see any acknowledgement to the source. Do the authors have permission to reproduce these images?
On line 60 the authors say: ‘After that, oil/water separation material with biomimetic super-wettability has undergone rapidly development.’ But they don’t give any examples. Some references should be added here for example Macromolecular Materials and Engineering 301 (9), 1032-1036.
On line 61 the authors say: ‘However, the short penetration channel and the relatively weak intrusion pressure can be easily broken by the high flux causing the it fails in separating oil/water mixtures [28]’ This problem has been solved by using 2 antagonistic meshes as described in ACS applied materials & interfaces 7 (34), 18915-18919. This is an important publication which the authors have missed. It should be added as a reference to this sentence.
Section 2 – Theoretical background. This manuscript is a review and so will be read by many people who are learning about oil-water separation, so should be correct. This section is incorrect, the authors have used an interpretation of the processes occurring which is too simple. Specifically, they have used static contact angles and advancing contact angles when they should use advancing and receding contact angles. They also only use water-in-air contact angles whereas there are actually 8 different contact angles which are important in the use of oil-water separation devices. These are from the different combinations of the media air, water, and oil. This is explained in Macromolecular Materials and Engineering 301 (9), 1032-1036.
Fig. 6 is not from reference 55.
Author Response
Reviewer 2
Comments and Suggestions for Authors
Do we need another review of oil-water separation devices? I feel that there are already enough.
Generally the written English is understandable but there are many mistakes in the text – too many for the reviewer to go through the text and highlight them. I recommend either taking more time to ensure that the English is as good as possible, or using an English editing service.
Response: Thank you for your comments. We have polished the language of the whole manuscript and the changes have been marked in yellow.
Photos shown in the figure 1 are very nice and add to the review article. However, I don’t see any acknowledgement to the source. Do the authors have permission to reproduce these images?
Response: Thank you for pointing out this for us. The permission of the photos have been added in Figure 1.
On line 60 the authors say: ‘After that, oil/water separation material with biomimetic super-wettability has undergone rapidly development.’ But they don’t give any examples. Some references should be added here for example Macromolecular Materials and Engineering 301 (9), 1032-1036.
Response: Thank you for your comments. The reference has been added.
On line 61 the authors say: ‘However, the short penetration channel and the relatively weak intrusion pressure can be easily broken by the high flux causing the it fails in separating oil/water mixtures [28]’ This problem has been solved by using 2 antagonistic meshes as described in ACS applied materials & interfaces 7 (34), 18915-18919. This is an important publication which the authors have missed. It should be added as a reference to this sentence.
Response: Thank you for your comments. The reference has been added.
Section 2 – Theoretical background. This manuscript is a review and so will be read by many people who are learning about oil-water separation, so should be correct. This section is incorrect, the authors have used an interpretation of the processes occurring which is too simple. Specifically, they have used static contact angles and advancing contact angles when they should use advancing and receding contact angles. They also only use water-in-air contact angles whereas there are actually 8 different contact angles which are important in the use of oil-water separation devices. These are from the different combinations of the media air, water, and oil. This is explained in Macromolecular Materials and Engineering 301 (9), 1032-1036.
Response: Thank you for your comments. We have modified the theoretical background and the discussions have been made as your suggestion. The relevant references have been added and highlighted in blue.
Fig. 6 is not from reference 55.
Response: Thank you for your comments. The reference has been changed.

Round 2
Reviewer 1 Report
publish as it is.
Reviewer 2 Report
The authors have made several improvements to their manuscript and it is now suitable for publication.